

# Evolution of sequence traits of prion-like proteins linked to amyotrophic lateral sclerosis (ALS)

Jiayi Luo and Paul M. Harrison

Department of Biology, McGill University, Montreal, Quebec, Canada

## ABSTRACT

Prions are proteinaceous particles that can propagate an alternative conformation to further copies of the same protein. They have been described in mammals, fungi, bacteria and archaea. Furthermore, across diverse organisms from bacteria to eukaryotes, prion-like proteins that have similar sequence characters are evident. Such prion-like proteins have been linked to pathomechanisms of amyotrophic lateral sclerosis (ALS) in humans, in particular TDP43, FUS, TAF15, EWSR1 and hnRNPA2. Because of the desire to study human disease-linked proteins in model organisms, and to gain insights into the functionally important parts of these proteins and how they have changed across hundreds of millions of years of evolution, we analyzed how the sequence traits of these five proteins have evolved across eukaryotes, including plants and metazoa. We discover that the RNA-binding domain architecture of these proteins is deeply conserved since their emergence. Prion-like regions are also deeply and widely conserved since the origination of the protein families for FUS, TAF15 and EWSR1, and since the last common ancestor of metazoa for TDP43 and hnRNPA2. Prion-like composition is uncommon or weak in any plant orthologs observed, however in TDP43 many plant proteins have equivalent regions rich in other amino acids (namely glycine and tyrosine and/or serine) that may be linked to stress granule recruitment. Deeply conserved low-complexity domains are identified that likely have functional significance.

## BACKGROUND

Prions in eukaryotes have been linked to diseases, evolutionary capacitance, large-scale genetic control and long-term memory formation. Prion formation and propagation have been studied extensively, particularly in the model organism *Saccharomyces cerevisiae*, a budding yeast. *S. cerevisiae* has >200 prion-like proteins that tend to have N/Q-rich (asparagine/glutamine-rich) domains of the sort observed in >10 known prion-formers (*An, Fitzpatrick & Harrison, 2016*; *Harbi & Harrison, 2014*). In humans, prion-like proteins have been linked mechanistically to amyotrophic lateral sclerosis (ALS) and other neurological/neuromuscular disorders, in particular the RNA-binding proteins FUS, EWSR1, TAF15, TDP43 and hnRNPA2 (*Chen et al., 2019*; *Picchiarelli & Dupuis, 2020*; *Smethurst, Sidle & Hardy, 2015*). A schematic diagram of the domain structure of the

Corresponding author
Paul M. Harrison,
paul.harrison@mcgill.ca

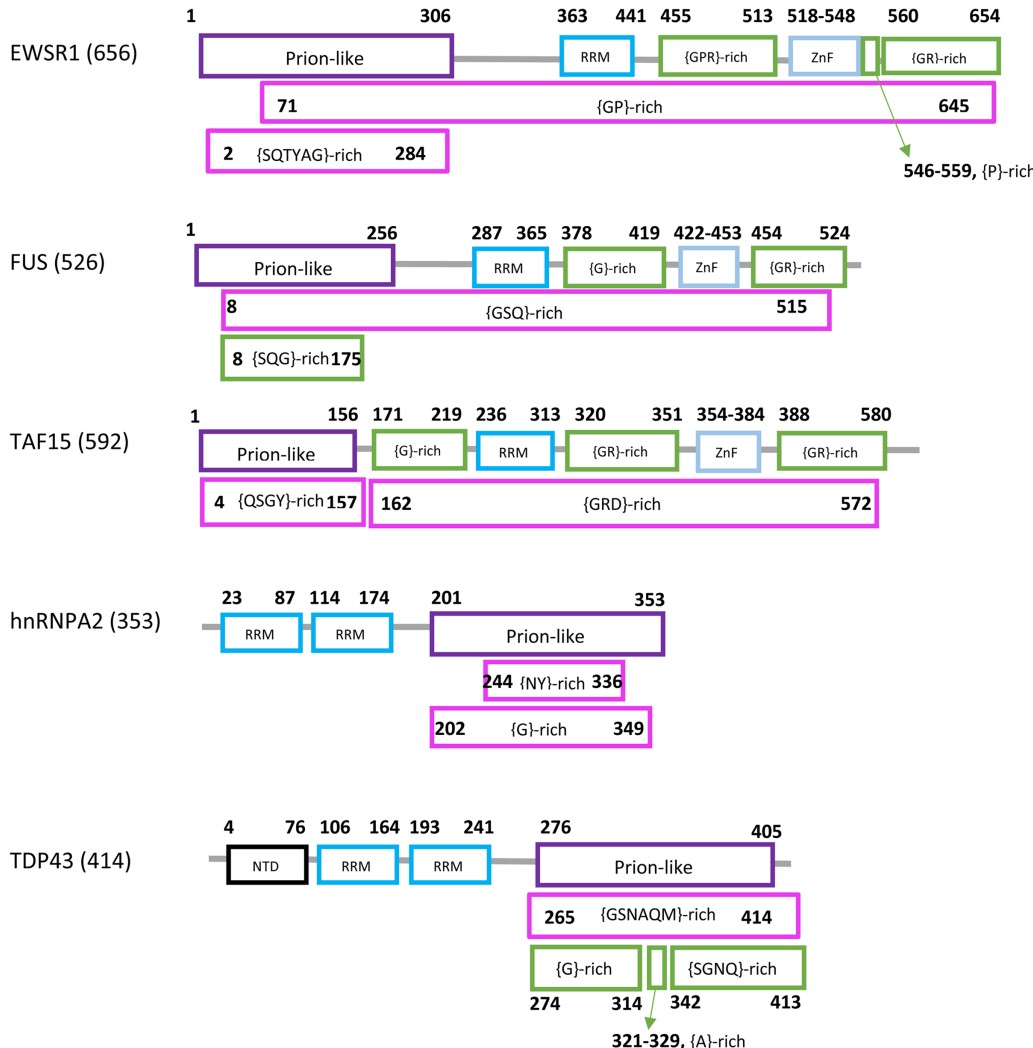

**Figure 1  Domain content of the human forms of EWSR1, FUS, TAF15, hnRNPA2 and TDP43.** The sequence names are at left with the sequence lengths in brackets. The sequences are represented by lines, with each domain as a box. The endpoints are labelled. Different types of domains are colour-coded. The endpoints of the prion-like regions (*purple*) are from PLAAC (*Lancaster et al., 2014*). The RRM (coloured *light blue*), ZnF (Zinc finger, *bright blue*) and NTD (N-terminal domain, *black*) assignments were determined as described in *Methods*. Compositionally-biased regions are labelled in the standard way according to fLPS output, {*xyz*}, meaning a bias for residues *x, y* and *z* in that order of precedence. They are coloured *magenta* if they were determined using default fLPS parameters except a threshold of $t = 1e{-}05$; they are coloured *green* if they are determined with fLPS parameters: $-m\ 5\ -M\ 25\ -t\ 1e{-}05\ -c$ *equal*. The latter parameter set is useful for labelling shorter 'low-complexity' regions. The data used to make this picture is available in tabular format in File S6.

human forms of these proteins is illustrated (Fig. 1). Each has a prion-like region and at least one RRM RNA-binding domain.

FUS is an RNA-binding protein involved in transcriptional activation and DNA repair (*Aman et al., 1996*; *Law, Cann & Hicks, 2006*; *Naumann et al., 2018*; *Wang et al., 2013*). Mutations in FUS are associated with ~5% of inherited ALS cases (*Monahan et al., 2018*); some of these mutations may cause intranuclear aggregation of FUS as part of its role in

pathomechanisms (*Nomura et al., 2014*). TDP43 has been shown to have multiple roles in repression of transcription, alternative splicing regulation and translational regulation (*Mitra et al., 2019*; *Ou et al., 1995*; *Sephton et al., 2011*). TDP43 is found in neuronal cytoplasmic aggregates in most ALS patients, although only a small minority of these have pathogenic mutations in the TDP43 gene (*Mackenzie et al., 2007*; *Sreedharan et al., 2008*). TDP43 has two close relatives, EWSR1 and TAF15. EWSR1 functions as a transcriptional activator (*Ohno, Rao & Reddy, 1993*), and is best known from its role in forming a chimeric oncoprotein linked to Ewing sarcoma and other tumors (*Anderson et al., 2018*). TAF15 is a TATA-binding protein that is a subunit of transcription factor IID (*Bertolotti et al., 1998*). TAF15 and EWSR1 have also been linked to neuronal aggregation and pathogenic mutations (*Couthouis et al., 2012*; *Jackrel & Shorter, 2014*). hnRNPA2/B1 functions diversely in mRNA transport, processing and metabolism (*Kim et al., 2013*; *Zhao et al., 2018*). It has been also found in neuronal cytoplasmic aggregates linked to ALS, and bears pathogenic mutations that are linked to both ALS and multi-system proteinopathy (*Kim et al., 2013*; *Zhao et al., 2018*).

Experiments using mutated sequences have demonstrated that the prion-like regions, low-complexity domains and intrinsically disordered parts of hnRNPA2, FUS and TDP43 proteins are linked to protein aggregation and phase separation into membraneless organelles, in particular stress granules (*Kim et al., 2013*; *Molliex et al., 2015*; *Patel et al., 2015*; *Wang et al., 2018*). *Molliex et al. (2015)* showed that low-complexity domains of hnRNPA2 mediate liquid-liquid phase separation and stress granule assembly. *Patel et al. (2015)* demonstrated that the physiological role of FUS depends on liquid-liquid phase separation and that disease-linked mutation promotes a liquid to solid phase transition. Different types of amino-acid residues appear to have distinct biophysical roles, for example in the liquid-liquid phase separation of FUS, tyrosine and arginine residues control the saturation concentration of phase separation, glycine enhances liquidity, and glutamine and serine promote the solidification of aggregates (*Wang et al., 2018*). In a recent study, it was observed that the tyrosine residues in the prion-like region in FUS are amongst the absolutely conserved residues across eukaryotes, which include also some of the glutamine, serine and glycine residues (*Dasmeh & Wagner, 2021*). TDP-43 lacks the arginine and glycine content seen in FUS and hnRNPA2, and thus the molecular determinants of phase separation differ; within the TDP-43 prion-like region there is a conserved tract that forms an alpha-helix that is important for phase separation in isolated prion-like subsequences (*Conicella et al., 2016*). Aromatic residues adjacent to glycines and serines also contribute to TDP-43 phase separation *in vitro* (*Li et al., 2018*). Many experiments have sought to use the budding yeast *Saccharomyces cerevisiae* as a model system for studying the aggregation of these proteins, or the aggregation of homologous or compositionally similar domains (*Monahan et al., 2018*).

Here, we have probed into the evolution across eukaryotes of the sequence traits of these proteins that are linked to ALS in humans, namely TDP43, FUS, TAF15, EWSR1 and hnRNPA2. Evolutionary trends are examined chiefly across plants and metazoans, since most of the data is available from these kingdoms. We discover that the RNA-binding domain architecture of these proteins is deeply conserved since their origination.

Prion-like regions are also prevalent since the origination of the protein families for FUS, TAF15 and EWSR1, and since the last common ancestor of metazoans for TDP43 and hnRNPA2. Prion-like composition is rare in any plant orthologs observed, however in TDP43 many plant proteins have equivalent regions rich in glycine and tyrosine and/or arginine, that may be linked to stress granule recruitment. Specific compositional biases predominate in many cases for these proteins across clades as ancient as the vertebrates, or even the metazoa. Shorter low-complexity regions can also be deeply conserved across kingdoms, indicating functional significance.

## METHODS

### Data

Sets of orthologs for the following protein families were downloaded from OrthoDB database version 10.1 at the Eukaryota level (*Kriventseva et al., 2019*): TDP43 (Group 1425837at2759), hnRNPA2 (Group 1202220at2759) and EWS/TAF15/FUS (Group 1539664at2759). These data also contain paralogs. For some analyses of the hnRNPA2 data, only proteins from a reduced list of representative eukaryotes used in a previous investigation were included (*An & Harrison, 2016*; *Su & Harrison, 2020*).

### Multiple sequence alignment and phylogenetic trees

Multiple sequence alignments were constructed using Clustal Omega (*Sievers & Higgins, 2018*), using default parameters. Phylogenetic trees were made using PhyML with Bayesian information criterion and aBayes branch support, and other parameters at defaults (*Guindon et al., 2005*). The aBayes branch support is a Bayesian-like transformation of the standard approximate likelihood ratio test implemented in PhyML, that is very fast and has high statistical power (*Anisimova et al., 2011*). Trees were saved in Newick format for input into the phylogenetic tree drawing tool Evolview version 2 (*Subramanian et al., 2019*). Annotations for prion-like regions, structured protein domains and compositionally biased regions (annotation described below) were formatted for input into Evolview using AWK scripts.

### Annotation of protein domains

From InterPro version 66.0, we extracted Pfam (version 34.0) protein domain positions (*El-Gebali et al., 2019*; *Mitchell et al., 2019*). These were checked against PROSITE and SMART domain annotations (*Letunic & Bork, 2018*; *Sigrist et al., 2010*), as in a previous study (*Su & Harrison, 2020*), and they diverge by <1% in the total number of domains labelled. Since there were some OrthoDB sequences that do not have Pfam annotations, we ran the Hmmsearch program using domain Hidden Markov Models (HMMs) downloaded from the Pfam website (http://pfam.xfam.org/) to label further domains (*El-Gebali et al., 2019*; *Mitchell et al., 2019*), with e-value threshold 0.01. Also, some further RRM RNA-binding domain annotations were made by searching for RRM domain protein structures in the PDB (Protein Data Bank) (*Bittrich et al., 2021*), and extracting the relevant protein domain sequences from the ASTRALSCOP database version 2.06 using PDB identifiers and residue ranges (*Fox, Brenner & Chandonia, 2014*). These sequences

were then compared to the proteomes using BLASTP 2.9.0 (*Altschul et al., 1997*) with e-value threshold 1e–04. All these annotations were then reduced for overlap by giving HMM-derived annotations precedence and otherwise sorting them in decreasing order of domain length and progressively flagging overlappers further down the lists for deletion.

### Prion-like regions, intrinsic disorder, and predicted stress-granule recruitment

Prion-like regions were annotated using the PLAAC program with default parameters (*Lancaster et al., 2014*). Experiments in *S. cerevisiae* wherein prion-forming sequences are scrambled yet maintain their prion nature have shown that prion-forming domains can be compositionally defined (*Ross, Baxa & Wickner, 2004*). PLAAC uses a Hidden Markov Model trained on known prion-forming domains from budding yeast that were detected through several tests for prion-like behaviour and amyloid formation (*Alberti et al., 2009*). Prion-like regions have compositions judged similar to those that can form prions. Although prion-like composition is typically identified as based on glutamine and asparagine bias, often regions that are biased for subsidiary residues of prion-like regions obtain high prion-like composition scores, *e.g.*, regions rich in tyrosine, glycine and serine. Prion-like regions were labelled if the PLAAC PRD prion-like composition score was >0.0 (as in a previous analysis (*Su & Harrison, 2019*)), and colour-coded according to PRD score in the annotated trees. Higher scores >15.0 are observed for known prion-forming domains (*Su & Harrison, 2019*). The program fLPS2 was used to annotate compositionally-biased (CB) regions (*Harrison, 2006*; *Harrison, 2017*; *Harrison, 2021*). Expected background frequencies = 0.05 were used in running fLPS2, with a threshold *P*-value of $10^{-6}$. CB regions were grouped together according to whether their bias signatures had a common form *{Xyzr….}*, where *X* is the main biasing residue and *yzr…*, *etc.* are subsidiary biases that can be in any order (*i.e.*, allowing for permutation of any subsidiary biases). The ten most common such CB regions are labelled on each phylogenetic tree. To analyze milder biases of experimental relevance, the '-z thorough' setting was used in fLPS2, with a threshold $P = 1 \times 10^{-4}$ (*Harrison, 2021*).

Total proportions of intrinsic disorder were calculated using IUPRED3 with default parameters (*Dosztanyi, 2018*; *Ward et al., 2004*), as in a previous work (*Su & Harrison, 2020*).

The server SGNN was used to estimate whether sequences can be recruited to stress granules (*Iglesias et al., 2021*). This tool was derived through analysis of data for the budding yeast *S. cerevisiae*, and thus may miss some of the compositional characteristics that drive stress granule recruitment in other organisms. Also, it may not work for regions of proteins that do not have prion-like characteristics, since it was trained to identify prion-like regions that are stress-granule recruited.

## RESULTS

Phylogenetic trees were calculated for eukaryotic protein families of TDP43, hnRNPA2 and EWSR1/TAF15/FUS. Protein domain content, prion-like composition, compositional biases, low-complexity regions, intrinsic disorder, sequence length and predicted stress

granule membership were calculated and annotated onto the trees (File S1). The data for making and annotating these trees is provided in File S2. Results are presented as follows:

   i)   trends in conservation of features since the last common ancestor of eukaryotes;
   ii)  distinct protein forms that are discovered in the metazoan and plant kingdoms;
  iii)  phylogenetic origins of FUS, TAF15 and EWSR1;
  iv)  specific evolutionary trends of sequence features for FUS, TAF15 and EWSR1.
   v)  trends in occurrences of shorter low-complexity regions (LCRs).

## Deep conservation of protein domains and prion-like regions since last common ancestor of eukaryotes

RNA-binding protein domain content is deeply conserved since the last common ancestor of eukaryotes for each of the protein families examined. For the hnRNPA2 family, the same number of RRM RNA-binding domains is largely maintained since the last common ancestor (LCA) of eukaryotes (>88% of orthologs) (File S1; Fig. 2C). Across the TDP43 family, all plant species have an equivalent RRM domain where metazoans have the NTD (N-terminal domain) (File S1). The total number of domains in TDP43 (N-terminal domain and RRM RNA-binding domains) is also deeply conserved (over >88% of sequences) (Fig. 2A). This NTD N-terminal domain has been shown to potentially have either DNA- or RNA-binding ability (*Chang et al., 2012*; *Qin et al., 2014*). Over the EWSR1/TAF15/FUS tree, a single RRM domain is maintained across 95% of sequences (Fig. 2B).

The prion-like regions (as defined by PLAAC) are generally conserved since the LCA of metazoans for all three protein families studied (File S1), with the percentage of metazoan orthologs maintaining some prion-like composition 74% for TDP43, 79% for hnRNPA2 and 90% for EWSR1/TAF15/FUS. The most notable exceptions are for TDP43 in some specific metazoan clades (File S1).

## Distinct protein forms in metazoans and plants

For the protein families that have substantial involvement in both plants and metazoans (*i.e.*, TDP43 and hnRNPA2), we can examine the distinct sequence characteristics in each of these kingdoms. These are summarized in Fig. 3. Most notably, the orthologs in plants have less intrinsic disorder and fewer prion-like regions with weaker prion-like compositions as defined using the PLAAC program (details in Fig. 2 and File S1). Interestingly however, the plant proteins are predicted to be more recruited to stress granules by the SGNN tool (details in Table 1). Zoom-ins of the plant and vertebrate parts of the TDP43 tree are shown in Fig. 4, to highlight these trends.

For TDP43 specifically, plant sequences do not at all assign to the NTD domain, but instead tend to a RRM domain assignment at the equivalent sequence position (File S1). Although they generally lack a prion-like region (with just a handful of exceptions; File S1), many of them (28%, 43/151) have a {GRY}-/{GY}-/{GS}-biased region instead at the same relative position in their sequences, and all but one of these is predicted to be recruited to

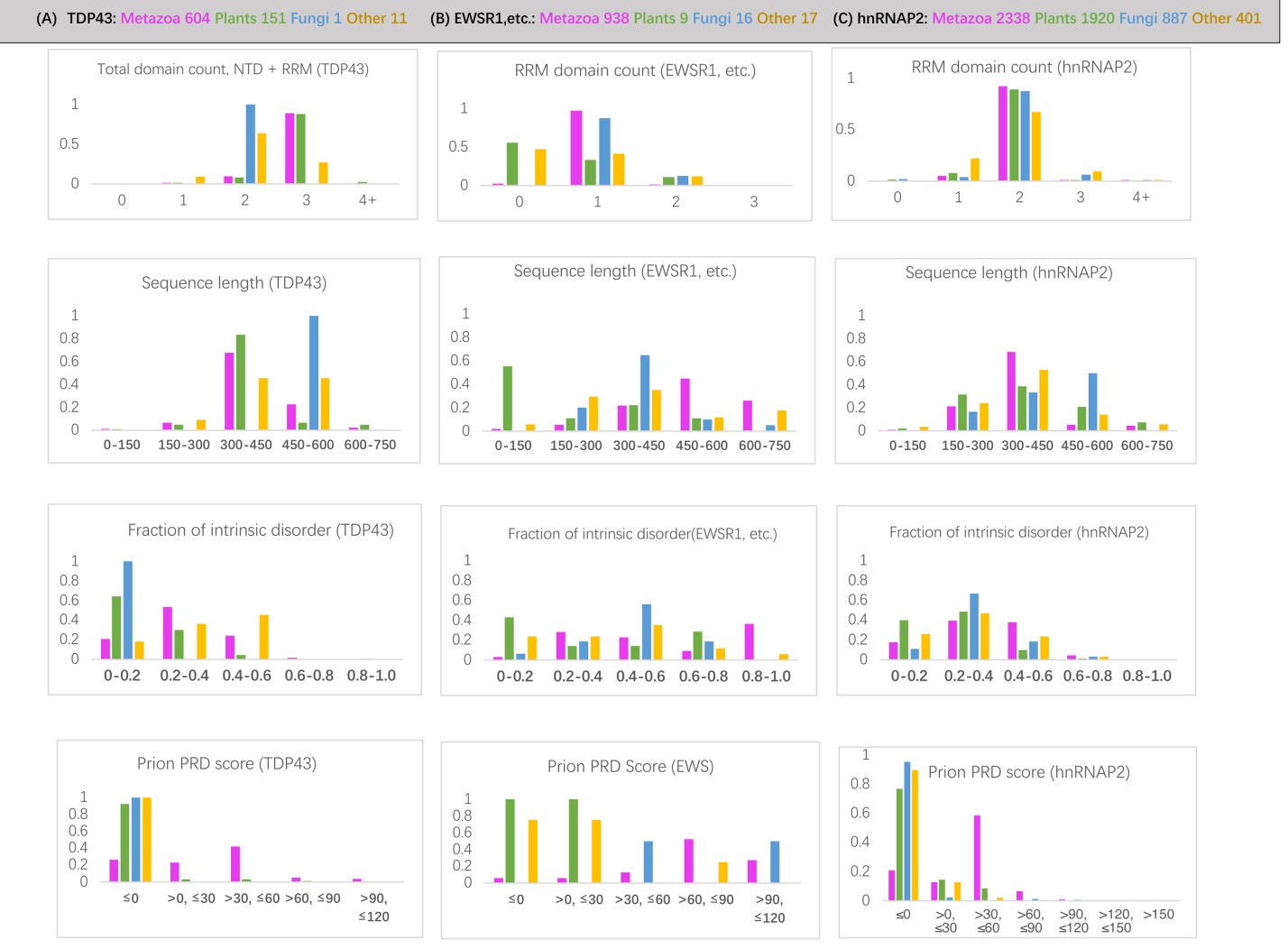

**Figure 2  Bar charts for the three sets of orthologs analyzed for various attributes.** They are arrayed in columns (A–C) for (total protein in brackets): TDP43 (763), EWSR1/TAF15/FUS (979), and hnRNPA2 (5503). The taxonomic groups are coloured as follows: magenta for metazoan, green for plants, blue for fungi and grey for others. The attributes analyzed (from top to bottom in each column): (i) *RRM RNA-binding domain count* for EWSR1/TAF15/FUS and total domain count for TDP43; (ii) *sequence length;* (iii) *proportion of intrinsic disorder*, as determined using the program IUPred3 (*Erdos, Pajkos & Dosztanyi, 2021*); (iv) *prion-like composition score* (PRD score) from the program PLAAC (*Lancaster et al., 2014*) for labelling regions of prion-like composition.                                             

stress granules by the SGNN tool (*Iglesias et al., 2021*). A {GMNQS}-rich prion-like region is predominant across vertebrates, with the exception of some fish clades (File S1, region of labelled by star symbols). Other animals demonstrate diverse bias patterns that are conserved in a clade-specific manner. One prominent feature of the zoom-ins of both the plant and vertebrate tree areas is a short ~30-residue {G}- or {Gx}-rich low-complexity region within the annotated prion-like regions, indicating a possible functional significance for such a domain (Fig. 4).

Similar trends are observed in the hnRNPA2 tree (File S1). A {GNY}-, {GNSY}-, or {GY}-rich prion-like region predominates in metazoans. However, few biased regions are

**TDP43**                    **hnRNPA2**

METAZOA

| **TOTAL   604** | |
|---|---|
| Total domain number  # | **2.88(±0.37)** |
| Prion-like PLAAC LLR score | **18.30(±10.22)** |
| PRD score | **29.76(±24.89)** |
| Intrinsic disorder | **0.33(±0.14)** |
| Sequence length | **418(±85)** |
| SGNN stress-granule membership predictions | **484 (80.1%)** |

| **TOTAL   2344** | |
|---|---|
| RNA-binding domain  # | **2.02(±0.30)** |
| Prion-like PLAAC LLR score | **24.45(±7.78)** |
| PRD score | **35.53(±23.73)** |
| Intrinsic disorder | **0.36(±0.16)** |
| Sequence length | **342(±106)** |
| SGNN stress-granule membership predictions | **537 (22.9%)** |

PLANTS

| **TOTAL   151** | |
|---|---|
| Total domain number  # | **2.92(±0.39)** |
| Prion-like PLAAC LLR score | **2.21(±5.33)** |
| PRD score | **3.24(±12.93)** |
| Intrinsic disorder | **0.18(±0.12)** |
| Sequence length | **383(±86)** |
| SGNN stress-granule membership predictions | **132 (87.4%)** |

| **TOTAL   1926** | |
|---|---|
| RNA-binding domain  # | **1.96(±0.30)** |
| Prion-like PLAAC LLR score | **13.78(±7.68)** |
| PRD score | **7.51(±13.5)** |
| Intrinsic disorder | **0.24(±0.13)** |
| Sequence length | **377(±138)** |
| SGNN stress-granule membership predictions | **1210 (62.8%)** |

**Figure 3 Summary of the trends for TDP43 and hnRNPA2 comparing plants to metazoa.** The total numbers of sequences are labelled at the top of each section. The first five attributes are listed as mean ± standard deviation, except the last one, which has a percentage of the total number of sequences in brackets. The attributes are: (i) count of RRM RNA-binding domains for hnRNPA2 and total domains for TDP43 (N-terminal domain plus RRM domains); (ii) LLR (log-likelihood) score and PRD score from the PLAAC program (*Lancaster et al., 2014*); (iii) proportion of intrinsic disorder, as judged by IUPred3 (*Erdos, Pajkos & Dosztanyi, 2021*); (iv) sequence length; (v) count of positive predictions of stress granule membership by SGNN (*Iglesias et al., 2021*). The values are colour-coded according to the key used in colour-coding the tree annotations in File S1.

observed in plant orthologs where they lack a prion-like region, and they maintain low levels of annotated intrinsic disorder (File S1; Fig. 3).

## Origins of FUS, TAF15 and EWSR1

FUS likely became widely conserved in an ancestor of the *Bilateria*, i.e., animals with embryonic bilateral symmetry, although there are six orthologs in cnidarians, so the exact evolutionary timepoint of its emergence is difficult to discern, since there may have been some initial complex patterns of gene loss at such early evolutionary stages. In a similar manner, TAF15 and EWSR1 seem to have become conserved in an early ancestor of vertebrates, but there are a small number of diverse assigned orthologs outside of the vertebrate clade (18 non-vertebrates out of 250 total for TAF15, and 6 out of 280 for EWSR1), so here again the exact evolutionary timepoint of the emergence of these proteins

**Table 1  Predictions of stress granule membership by SGNN.**

| Protein family | Range (Number of proteins) | SGNN prediction of stress granule membership | |
| --- | --- | --- | --- |
| | | Yes | No |
| EWSR1 | Eukaryota (280) | 4.6% | 95.4% |
| FUS | Eukaryota (396) | 13.6% | 86.4% |
| TAF15 | Eukaryota (250) | 21.6% | 78.4% |
| | | | |
| TDP43 | Eukaryota (763) | 81.7% | 18.3% |
| TDP43 | Metazoa (587) | 79.9% | 20.1% |
| TDP43 | Plants (151) | 87.4% | 12.6% |
| TDP43 | Fungi & Other (12) | 75.0% | 25.0% |
| | | | |
| hnRNPA2 | Eukaryota (5,503) | 38.9% | 61.1% |
| hnRNPA2 | Metazoa (2,338) | 23.0% | 77.0% |
| hnRNPA2 | Plants (1,920) | 63.0% | 37.0% |
| hnRNPA2 | Fungi (887) | 22.4% | 77.5% |
| hnRNPA2 | Other (410) | 52.9% | 47.1% |

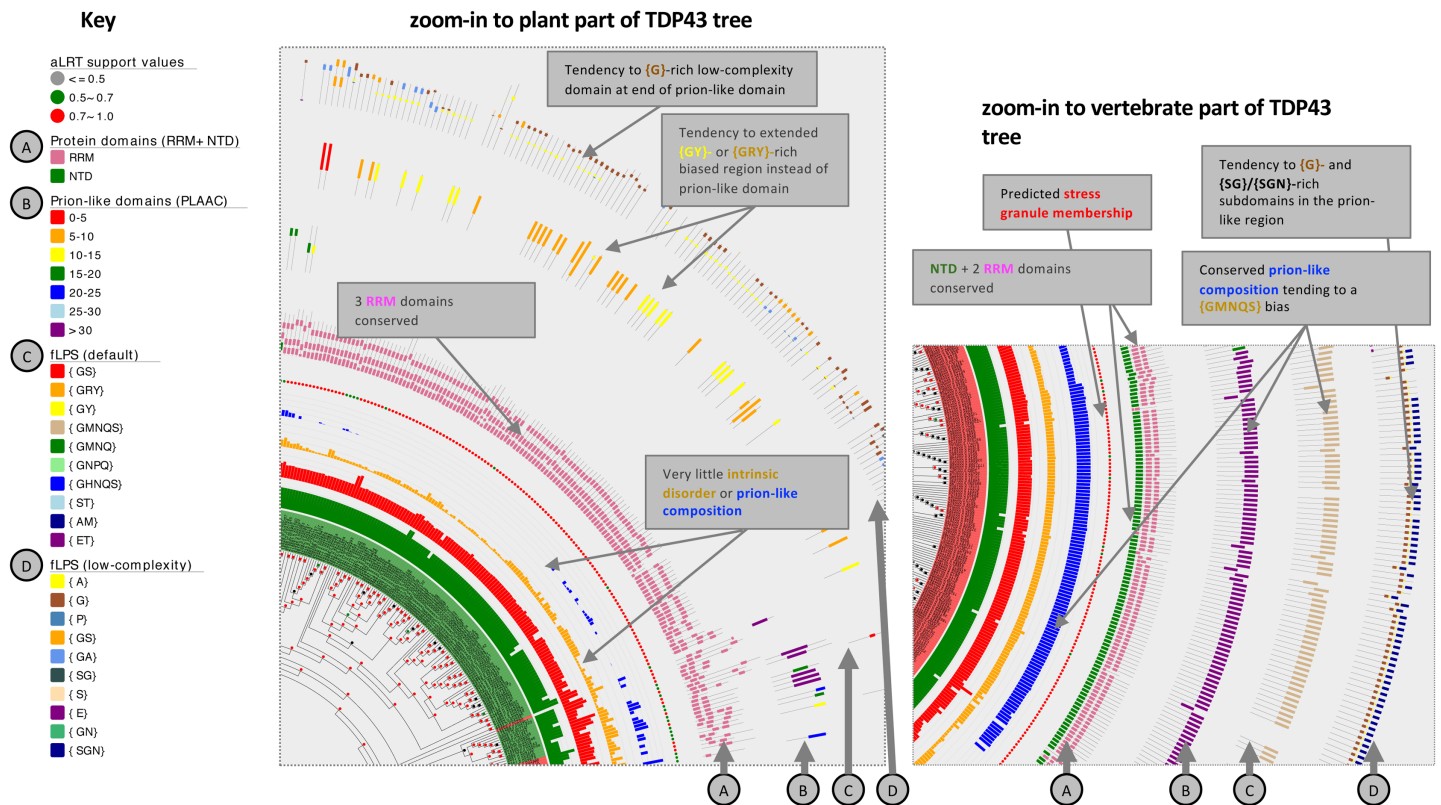

**Figure 4  Zoom-ins of the TDP43 phylogenetic tree illustrating the detail of the plant and vertebrate ortholog annotations.** The key for tree attributes is at left and is described in detail in File S1. The trees are drawn with Evolview version 2 (*Subramanian et al., 2019*). Salient distinguishing features of either tree zoom-in are labelled.

(a)

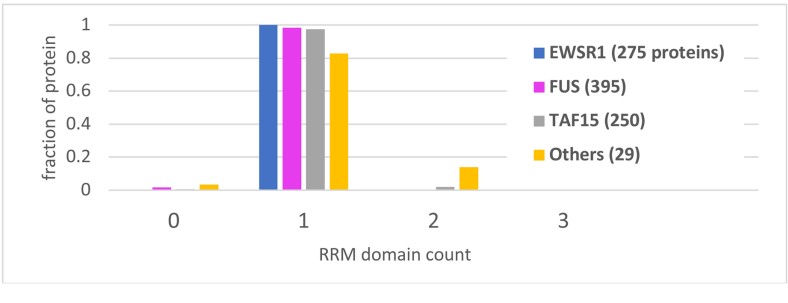

(b)

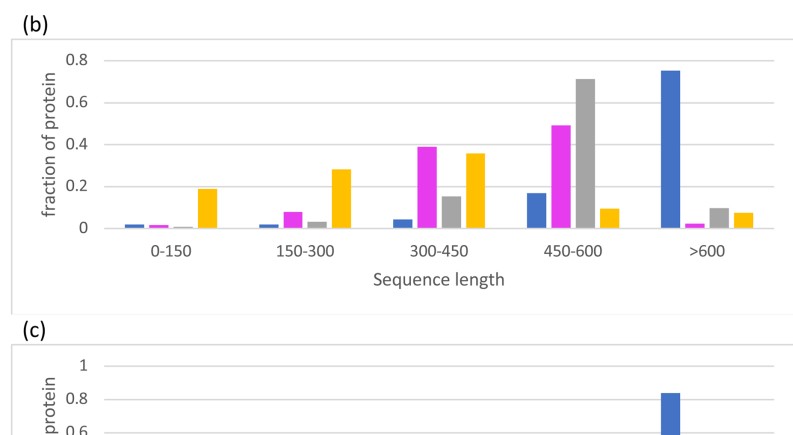

(c)

(d)

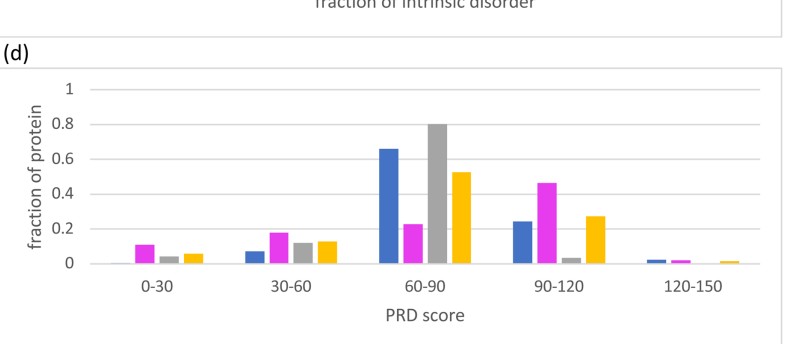

**Figure 5 Bar charts for the tree of EWSR1/TAF15/FUS orthologs showing the distributions of various attributes for each of the proteins EWSR1, TAF15 and FUS.** The total number of sequences for each of the proteins (plus 'others') are in the key at the top of the figure. The y-axis is the fraction of proportion of proteins. The figures panels are for: (A) RRM RNA-binding domain count; (B) sequence length; (C) proportion of intrinsic disorder, as determined using the program IUPred3 (*Erdos, Pajkos & Dosztanyi, 2021*); (D) PRD score from the program PLAAC (*Lancaster et al., 2014*) for labelling regions of prion-like composition.

is also difficult to discern. Generally, the small number of non-metazoan homologs in the tree are much shorter sequences (particularly in plants), and correspondingly have less intrinsic disorder (Fig. 5).

**Table 2  Zinc finger domains in the EWSR1/TAF15/FUS family.**

| Protein family | Number of Zn-finger domains | | | |
| --- | --- | --- | --- | --- |
| | 0 | 1 | 2 | 3+ |
| EWSR1 | 27 (10%) | 193 (69%) | 59 (21%) | 0 |
| FUS | 27 (7%) | 349 (88%) | 21 (5%) | 0 |
| TAF15 | 12 (5%) | 235 (94%) | 2 (1%) | 0 |

## Specific evolutionary trends for EWSR1, TAF15 and FUS

Apart from the deep conservation of RNA-binding domains and prion-like regions (Figs. 2 and 5), the individual sequence evolution of EWSR1, TAF15 and FUS has unfolded idiosyncratically.

In particular, each of these protein families have characteristic amounts of deeply conserved intrinsic disorder (Fig. 5C). The EWSR1 family has the highest proportions of annotated intrinsic disorder, typically >0.8, that have been conserved across clades since the last common ancestral sequence (Fig. 5C, File S1). Comparatively, TAF15 has intermediate levels of intrinsic disorder (mode 0.4–0.6), with the lowest for FUS proteins (mode 0.2–0.4) (Fig. 5C). These tendencies do not correlate with PLAAC prion-like composition scores, which are notably higher for FUS compared to the other two proteins (Fig. 5D). All three families maintain some degree of prion-like compositions (score >0) over the vast majority of the family members (97% of members for EWSR1, 93% for TAF15 and 85% for FUS).

A single Zn-finger domain is a deeply conserved component of both the FUS and TAF15 protein families (File S1; Table 2). However, EWSR1 has greater variation in the number of Zinc-finger domains compared to the other two protein families (Table 2), with one in five orthologs having two domains, and one in 10 having none.

The default fLPS program was used to detect longer compositionally biased regions (CBRs). We discovered that CBRs are maintained across deep and diverse clades when we examined the detail of the annotated trees (File S1). TAF15 conserves a {GR}- or {GY}-rich bias across mammals (bias $P$-values $< 1 \times 10^{-10}$) with the C-terminal R bias maintained in other vertebrates (labelled as a {CR} bias encompassing also the Zn-finger domains, $P$-values $< 1 \times 10^{-6}$). Where a predominant {GY} bias is labelled, it stretches across the whole sequence, whereas the {GR} bias is beyond the Zn-finger domains at the C-terminus. For FUS, the fLPS2 program assigns a biased region across most of the sequence that varies between {GQSY} or {GQRSY} in a clade-specific manner (bias $P$-value $< 1 \times 10^{-24}$). In EWSR1, a very specific {GMPRSTY} bias predominates across all vertebrates (bias $P$-value $< 1 \times 10^{-52}$), with the exception of fish, which conserve a similar, but also very specific {GMPQRSTY} bias ($P$-value $< 1 \times 10^{-46}$) (File S1).

We also examined for milder compositional biases of experimental relevance, i.e., involved in recruitment to stress granules, such as {R} and {Y} bias (fLPS2 $P$-value $\leq 1 \times 10^{-4}$) (*Wang et al., 2018*). Both {R}- and {Y}-biased regions occur with high frequency, but also other positively-charged regions, i.e., {K}-biased regions, are quite common, particularly in TAF15 orthologs (File S3). {R}-biased regions are characteristically shorter
than {Y}-biased regions (File S4), indicating that arginine residues corresponding to tyrosine residues that might interact with them are less dispersed along the sequence. The most prevalent bias across all protein types is {G}.

### Low-complexity regions

We analyzed the distribution of low-complexity regions (LCRs) annotated using the program fLPS 2.0 and parameters recommended for shorter LCR discovery (*Harrison, 2021*). LCRs associated with RNA-binding, such as the {GR}-/{GPR}-/{G}-rich regions seen in EWSR, TAF15 and FUS, are sometimes termed RGG or RG regions, and associated functionally with RNA-binding (*Chong, Vernon & Forman-Kay, 2018*). In addition to these regions, we discovered that other LCRs are also deeply conserved across kingdoms for each of three protein families examined (File S5). For example, a short {P}-rich tract (as exemplified in EWSR1, Fig. 1) is deeply conserved across EWSR1 orthologs (219/249 cases) and some other non-metazoan orthologs (20/249 cases) (14.3 ± 5.4 residues in length, median *P*-value = 8.6e−08). A short {A}-rich LCR is deeply conserved across many clades for the TDP43 family, specifically within the putative prion-like domain in vertebrates (11.7 ± 8.9 residues in length, median *P*-value = 1.1e−06, visible in Fig. 4), and corresponds to an alpha-helical region that likely functions in phase separation (*Conicella et al., 2016*). In general, the top ten LCRs are conserved across kingdoms with few exceptions (non-grey bars in the charts, File S5).

## DISCUSSION

Since they emerged evolutionarily, the proteins of these five families have had deeply conserved RNA-binding capability, that varies only rarely in terms of RNA-binding domain number, indicating selection pressure against further RNA-binding domain duplication. In tandem across metazoa, the TDP43 and hnRNPA2 families demonstrate a selection pressure to maintain prion-like composition, in contrast to the situation in the plants, where prion-like character is markedly rarer, particularly for TDP43 plant orthologs. However, such proteins may still function in biomolecular condensate formation, particularly plant stress-granule formation in response to environmental cues (*Cuevas-Velazquez & Dinneny, 2018*; *Maruri-Lopez et al., 2021*). The most common compositional biases in plant TDP43 orthologs are a group of {GRY}/{GY}/{GS} biases that may be linked to stress granule recruitment, indeed these proteins are almost universally predicted to be recruited to stress granules. Such subsidiary biases for G, Y or S are sometimes observed in prion-forming domains (*Lancaster et al., 2014*). Mutational experiments on FUS have shown that separate regions of arginine and tyrosine bias interact with each other through specific side-chain arginine-tyrosine bonding to control the saturation concentration at which liquid-liquid phase separation occurs, with glycine residues shown to be necessary to maintain membraneless organelle liquidity (*Portz, Lee & Shorter, 2021*; *Wang et al., 2018*). Tyrosine-tyrosine interactions also contribute to phase separation (*Sun et al., 2011*). In general, however the five protein families examined here do not demonstrate the very deep eukaryote-wide conservation of both RNA-binding domain architecture and prion-like composition observed for the TIA-1 protein family

(*An & Harrison, 2016*; *Su & Harrison, 2020*). TIA-1 has been shown to form prion-like protein aggregates in both its yeast and human forms and is also linked mechanistically to ALS, and through rare inherited mutations in some human cohorts (*Gu et al., 2018*; *Zhang et al., 2018*).

A caveat here is that the SGNN tool used to estimate stress granule recruitment was only trained on *S. cerevisiae* proteins with prion-like composition, so that it is unclear what results for proteins with weak prion-like composition, especially in divergent eukaryotes, might mean. Also, it is important to bear in mind that the PLAAC program for identification of prion-like composition was trained on data obtained from experiments on *S. cerevisiae* proteins. However, there has been some success in using the PLAAC program to identify prion-forming domains in other eukaryotes, and in bacteria and archaea (*Chakrabortee et al., 2016*; *Harrison, 2019*; *Kim et al., 2013*; *Sideri et al., 2017*; *Yuan & Hochschild, 2017*; *Zajkowski et al., 2021*), but there may be other prion-forming domain compositions that are not sampled during the evolution of budding yeasts (*An, Fitzpatrick & Harrison, 2016*; *Wang & Harrison, 2021*).

In general, compositional biases in these protein families vary diversely from clade to clade, but can be deeply conserved within a clade, *e.g.*, a specific {GMNQS} bias conserved in TDP43 across vertebrates (with the exception of fish). This mode of evolution suggests periodic rare shifts in the molecular grammar governing the functional traits of these sequences, possibly linked to encoding of prion formation or stress granule recruitment. The maintenance of prion-like character despite underlying shifts in compositional bias is also a feature of the evolution of prion-forming domains of the budding yeast *Saccharomyces cerevisiae* (*Su & Harrison, 2019*). These ALS-linked ortholog sequences could thus be further analyzed mutationally to dissect how the side-chain interactions governing possible prion formation or stress granule recruitment capability may have changed at key time points of eukaryotic organismal evolution. Shorter low-complexity tracts that are conserved across diverse eukaryotic clades (such as the {P}-rich example that was highlighted) could also be dissected to discern their functional significance.

## CONCLUSIONS

We have presented a detailed analysis of the evolution of sequence traits of five ALS-linked protein families. We discovered that the RNA-binding domain architecture of these proteins is deeply conserved since the origination of these proteins, or since the last common ancestor of metazoa. Prion-like regions are also prevalent since their origination for FUS, TAF15 and EWSR1, and since the last common ancestor of metazoa for TDP43 and hnRNPA2. We discussed how the plant orthologs of TDP43 and hnRNPA2 are distinct from their metazoan counterparts, but still often have features that can be linked to stress granule formation. These data are useful in developing further hypotheses about these ALS-linked protein families that can be experimentally tested, particularly when assessing how corresponding protein sequences have changed in model organisms. Specific conserved subdomains and low-complexity regions that were observed could be examined experimentally for their functional significance. This study demonstrates the

utility of combined application of diverse sequence annotation programs to characterize evolutionary trends.

### Funding
This work was supported by the Natural Sciences and Engineering Research Council of Canada. The funders had no role in study design, data collection and analysis, decision to publish, or preparation of the manuscript.

### Grant Disclosures
The following grant information was disclosed by the authors:
Natural Sciences and Engineering Research Council of Canada.

### Competing Interests
The authors declare that they have no competing interests.

### Author Contributions
- Jiayi Luo performed the experiments, analyzed the data, prepared figures and/or tables, authored or reviewed drafts of the article, and approved the final draft.
- Paul M. Harrison conceived and designed the experiments, performed the experiments, analyzed the data, prepared figures and/or tables, authored or reviewed drafts of the article, and approved the final draft.

### Data Availability
The raw data is available in the Supplemental Files.

### Supplemental Information
Supplemental information for this article can be found online at http://dx.doi.org/10.7717/peerj.14417#supplemental-information.

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
