# Peer review of "Evolution of sequence traits of prion-like proteins linked to amyotrophic lateral sclerosis (ALS)"

_PeerJ, doi:10.7717/peerj.14417_

## Round 0.1 · original submission · Major Revisions

The three Reviewers raise valid issues with respect to the presentation that should be carefully addressed in a revised version. The manuscript requires clarifications, more detailed information, more context and discussion including relevant references, and availability of raw data.

Reviewer 1 ·

Basic reporting

The study describes phylogenetic comparisons for five RNA-binding proteins with prion-like domains. Such an in-depth comparison of domain and sequence properties and how those protein families have evolved could be of interest, however the study is currently not well presented and lacks clarity.

The introduction briefly introduces the context and the five proteins selected for study. Here, more background information should be given on those proteins and the biological context explained for the non-expert reader. For example, the differences between TDP43 and “its close relatives” (?) TAF15 and EWSR1 need to be described. It may be helpful for the reader to add schematic representation of the human proteins with location of different domains (domain structure).

The abstract and introduction give the impression that yeast/fungi data would also be discussed thoroughly. However, this is not the case as most comparisons are for plants and metazoan. This needs to be reframed.

The results should be written in a more reader-friendly fashion. Currently, it jumps from one topic to the other, missing a connecting line that guides the reader and explains the rationale adequately (e.g, why has the length of the proteins been compared?).

The quality of Fig. S1 is good but quality of main figures needs to be improved. The discussion mentions a Figure SUMMARY (line 214). I was not clear what this refers to.

Importantly, the raw data (output files) must be made available in tabular format (txt/ .xsl file). This should include data used to create the Supplemental Fig. S1.

Experimental design

This is a bioinformatics study that extracted information from different databases. The date and database version needs therefore to be specified.
The Materials and Methods section needs more details on actual parameter setting for data retrieval and generating the MLS (line 89, 93).

Validity of the findings

The findings seem valid though leave room for interpretation. In this regard, discussion of alternative interpretations as domains could have been lost, mutated and/or replaced and evolutionary context including fungi would be of interest.
Sentences like “TAF seems to have become conserved in an early ancestor of vertebrate” (line 161) are vague and need more detailed specification and data to be understood. Likewise, the sentence starting at line 223 is long and hard to understand.

Additional comments

The study could be of interest but is currently not well presented and hard to follow. The discussion should bring the findings into larger context and cover fungi, metazoan, plants and others (what are these?). Raw data needs to be added.

Reviewer 2 ·

Basic reporting

This article is clearly written and easy to understand. However, there were a few minor issues that should be addressed to help clarify the manuscript:

1) Some basic definitions might be helpful for readers. The authors start by saying that yeast have >200 prion-like proteins. It would be helpful to explain exactly how the authors are defining prion-like proteins. The authors only vaguely state that these proteins have N/Q rich domains "of the sort observed" in known prions, but prion-like composition goes beyond just Q and N; indeed, many of the prion-like domains discussed in the article have far lower N/Q content than the known yeast prions.

2) Some of the reference choices seem a little arbitrary, and don't reflect the foundational work that supports the accompanying statements. For example, in lines 58-59, the authors discuss the role of prion-like domains from hnRNPA2, FUS, and TDP-43 in aggregation, phase separation, and membraneless organelles, but their two citations only address a tiny portion of this statement. Kim et al is a reasonable citation for aggregation of hnRNPA2, but the foundational papers involving the role of prion-like domains in FUS and TDP-43 aggregation are not cited. Wang et al does address phase separation (by FUS family proteins), but the earlier papers actually demonstrating phase separation by these proteins (Molliex et al, Patel et al) would be more appropriate here.

3. The authors have a tendency to cite themselves, when earlier foundational work from other labs would be more appropriate. For example, in line 38, they cite two of their own papers to argue that yeast have >200 prion-like proteins, rather than the seminal work of Alberti et al (or even Michelitsch & Weissman) that originally showed this.

Experimental design

The experimental design is appropriate, although as discussed below, some analysis of the limitations of each method (PLAAC, SGNN, etc.) might be helpful.

Validity of the findings

The underlying is appropriate and the data robust. However, there are a few issues that could be addressed to strengthen the conclusions.

1) It would be helpful to provide more context about what is already known (or has been proposed) regarding the sequence/compositional features responsible for the various activities of each protein. For example, when discussing the compositional biases in TDP43 and their possible role in stress granule recruitment (lines 217-219), the authors cite one study that proposed R-Y interactions are critical for FUS phase separation. However, this is only one of the types of interactions that has been proposed for FUS. There has been extensive work for each of these proteins examining various interactions that promote phase separation, and these features seem to differ between proteins (see for example PMID 33446423, which discusses the different features driving phase separation for TDP43 versus FUS). Some context about what is already known is critical for interpreting the current experiments.

2. The authors should include some discussion about the source and limitations of the different analysis tools that they use (PLAAC, SGNN, etc.). For PLAAC, as discussed above some context as to what it means to have prion-like composition would be helpful. Likewise, a brief discussion of the evidence that supports a focus on composition would be helpful. For SGNN, it is important to note that this tool was developed based on a prior study in yeast, and thus may not reflect the compositional characteristics that drive stress granule recruitment in other organisms (yeast stress granules have distinct physical properties from the stress granules observed in higher eukaryotes). Likewise, it is important to note that this tool was only validated on proteins with prion-like composition; this is a critical caveat, for example in lines 142-146, where the authors argue that many plant proteins lack prion-like domains, yet are predicted to go to stress granules.

Additional comments

A couple very minor points:
1. In line 37, the authors describe "N/Q-rich (glutamine/asparagine-rich)" domains. This should either read "N/Q-rich (asparagine/glutamine-rich)" or "Q/N-rich (glutamine/asparagine-rich)."

2. In Figure 2, there are a couple places where the authors have standard deviations indicated with just a "+" instead of a "+/-" (the metazoa and plant PRD scores). The authors also indicate that the items in this table are mean +/- SD, but the last one is not, so the meaning of the % in the parenthesis should be explicitly defined.

Reviewer 3 ·

Basic reporting

See comments 2,5,6,7 in section 3.

Experimental design

See comments in section 3.

Validity of the findings

Review of "Evolution of sequence traits of prion-like proteins linked to ALS"

Several RNA-binding proteins that have been linked to ALS in humans have domains with amino acid compositional biases similar to those found in yeast prions (prion-like domains). Such domains tend to be disordered and can be involved with phase-separation and stress-granule recruitment. In the present work the authors examine, for each of these the five proteins, the conservation of the domain structure, compositional biases, prion-like domains, predicted disorder and predicted recruitment to stress granule across species.

They find that many of these attributes are well-conserved across metazoan species but with some domains absent or altered in plants. They authors present this as as indication of selective pressure to maintain these properties, though this is an informal argument rather than one based on a formal evolutionary model.

Summaries of conservation take the form of (1) barplots showing the distribution across homologs of scores for each feature, broken into five of so bins, without reference to the evolutionary relationships between the homologs; (2) a display of these scores in the context of phylogenetic tree (full trees in supplement, details in main text).

Major issues:

(1) It is not always clear whether the result indicate (likely) properties of the proteins or just biases in the algorithms due to a mismatch between training data and test data.

(1a) For example, line 142 says "Most notably, the orthologs in plants have less intrinsic disorder and fewer prion-like domains with weaker prion-like compositions. Interestingly however, the plant proteins are predicted to be more recruited to stress granules by the SGNN tool". The authors say that this suggests "periodic rare shifts in the molecular grammar governing stress granule recruitment" (line 231). But the cited ref for the SGNN tool by Iglesias et al. says "of note, SGnn was trained on top of pre-defined yeast PrLDs and, accordingly, it is not intended to be used in full-length proteins or regions that do not possess prion-like features". How does this impact the interpretation of the findings?

(1b) There may be similar considerations related to background AA frequencies used by other tools, and potential consequences of this should be discussed.

(2) The results are not provided in a form that allows re-use; for this, supplemental tables of per-protein features would be more helpful that the supplemental plots of annotated phylogenetic trees.

Medium issues:

(3) Lines 10-105 say "Since there were some OrthoDB sequences that do not have Pfam annotations, and other annotations may be incomplete, we generated additional protein domain annotations. Sequence fragments for protein domain annotations were extracted and mapped onto other protein sequences using BLASTP with e-value threshold 1e-04. These were then reduced for overlap by sorting them in decreasing order of domain length and progressively flagging overlappers further down the list for deletion." Were these merged with Pfam annotations (for those sequences that already had them) or were existing Pfam annotations ignored? Why not just use Pfam-scan or similar to annotate Pfam domains, rather than matching to a single reference domain with BLASTP? Aren't Pfam domains represented by HMMs, which can give more weight to more important positions in the domains – information that could be lost with BLASTP matching?

(4) Line 120 says "The total percentage of intrinsic disorder was calculated for each whole protein sequence, since there is not an intrinsic disorder score for a specific tract that can be extracted from the output." What tracts would the authors have scored otherwise? The IUPRED3 server returns per-residue scores for single sequence queries, which can be subsetted as desired. I don't know if it does that for multi-sequence queries but there is an API for automating many single-sequence queries, and there is also a downloadable version which may provide per-residue scores.

Minor issues:

(5) First two sentences say "Prions are proteinaceous particles..." and "...>10 of these proteins domains...", so "these" in the 2nd sentence should be referring to proteins rather than protein domains.

(6) The annotations of these scores on the trees include some unfortunate scale breaks (e.g.: 0, 0.2, 0.4, 0.6000000000000001, 0.8)

(7) Line 204 says "indicating that arginine residues corresponding to tyrosine residues that might interact with them are less dispersed along the sequence" but the context for this is not introduced until the next section, in line 220: "Mutational experiments on FUS have shown that separate regions of arginine and tyrosine bias interact with each other through..."

---

## Round 0.2 · Minor Revisions

Reviewer 3 highlights a few remaining minor issues that should be addressed.

Reviewer 2 ·

Basic reporting

The authors have appropriate addressed my concerns from the previous version. The current version is clearly written.

Experimental design

The experimental design is appropriate, and the authors have appropriately clarified the limitations of each method.

Validity of the findings

The findings are robust and statistically sound.

Reviewer 3 ·

Basic reporting

Review of revisions of "Evolution of sequence traits of prion-like proteins linked to ALS".

The authors have made an effort to address the issues I raised in my initial review. The MAJOR issue (1a/1b) - on the extent to which the findings reflect biases in algorithms related to training data and background AA frequencies (which was also noted by another reviewer) - is still a concern, but the authors have added a discussion of this limitation. This may be the best that can reasonably be expected here, given the absence of experimental data on prion-like behavior and stress granule composition in many species.

Issue 3 has been addressed by using Hmmsearch on Pfam HMMs, although there is now additional text on lines 145-147 on "extracting the relevant protein domain sequences from the ASTRALSCOP database version 2.06 and comparing these domains to the proteomes using BLASTP 2.9.0". It is not clear that the "Annotation of protein domains" section as a whole is described in adequate detail to allow this part of the analysis be replicated, though perhaps if default parameters were used everywhere it may be. Regardless, this issue is mitigated in part by the supplemental tables that include domain annotations, so may be acceptable as is.

Several minor issues in newly-added text:

(NOTE: all lines numbers are for the "tracked changes" Word version, not the pdf version.)

Line 18: Grammar is a bit awkward – not a big deal, but this is the first sentence of the abstract. Change "protein, that have" to "protein. They have" ?

Line 65: "TDP43 has two close relatives EWSR1 and TAF15" seems like it's missing a comma after "relatives", though that may give the impression that those are the only two close relatives of TDP43. This sentence could be rephrased to make its intended meaning, and relation to next sentence, more clear. Would changing this sentence to "EWSR1 and TAF15 are close relatives of TDP43" work?

Line 81 and 83: Commas before "et al" aren't needed, are they?

Line 576: "annotated" should be "annotate"

Also:

Line 676: missing issue and page number for Harbi & Harrison ref.

Experimental design

NA

Validity of the findings

NA

Additional comments

NA

---

## Round 0.3 · accepted · Accept

The authors have adequately addressed the remaining issues raised.